# Inositol Derivatives with Anti-Inflammatory Activity from Leaves of *Solanum* *capsicoides* Allioni

**DOI:** 10.3390/molecules27186063

**Published:** 2022-09-16

**Authors:** Yan Liu, Xin Meng, Han Wang, Yan Sun, Si-Yi Wang, Yi-Kai Jiang, Adnan Mohammed Algradi, Anam Naseem, Hai-Xue Kuang, Bing-You Yang

**Affiliations:** Key Laboratory of Basic and Application Research of Beiyao, Heilongjiang University of Chinese Medicine, Ministry of Education, Harbin 150040, China

**Keywords:** inositols, anti-inflammatory activities, *Solanum capsicoides*, Solanaceae

## Abstract

Eight new inositol derivatives, solsurinositols A–H (**1**–**8**), were isolated from the 70% EtOH extract of the leaves of *Solanum capsicoides* Allioni. Careful isolation by silica gel column chromatography followed by preparative high-performance liquid chromatography (HPLC) allowed us to obtain analytically pure compounds **1**–**8**. They shared the same relative stereochemistry on the ring but have different acyl groups attached to various hydroxyl groups. This was the first time that inositol derivatives have been isolated from this plant. The chemical structures of compounds **1**–**8** were characterized by extensive 1D nuclear magnetic resonance (NMR) and 2D NMR and mass analyses. Meanwhile, the in vitro anti-inflammatory activity of all compounds was determined using lipopolysaccharide (LPS)-induced BV2 microglia, and among the isolates, compounds **5** (IC_50_ = 11.21 ± 0.14 µM) and **7** (IC_50_ = 14.5 ± 1.22 µM) were shown to have potential anti-inflammatory activity.

## 1. Introduction

Inositols are abundant in the brain and other mammalian tissues, and can be defined as polyols or cyclitols, having the basic structure of one cyclohexane with a hydroxy group bound to each carbon atom of the hexane ring [1]. The study showed that inositol, inositol lipids, and inositol phosphates have become universal constituents of eukaryotes [2]. Inositols and their derivatives were extracted and isolated from Asteraceae, Dioscoreaceae, Amaranthaceae, and Solanaceae, et al. in [3]. Another study showed that myo-inositol plays a positive function in follicular maturity in human follicular fluids and is a marker of good-quality oocytes. These findings prompted the idea that inositol plays an essential role in human reproduction [4]. In addition, myo-inositol dietary supplementation was shown to improve insulin sensitivity, which suggests that inositol plays a potential role in metabolism [5]. Therefore, it is necessary to study inositols and their derivatives.

*Solanum capsicoides* Allioni (*S. capsicoides*), one of the members of Solanaceae, has been used in traditional Chinese medicine. It has been recorded as being widely used in China, India, and South America in classical medical books. Modern pharmacological studies have shown that the aerial parts of *S. capsicoides* have antihypertensive and antidepressant-like effects [6,7]. Their extracts exhibited anti-inflammatory and analgesic effects in [8,9]. Currently, research on the chemical compositions of this plant has revealed the presence of withanolides, steroidal alkaloids, steroidal saponins, and phenylpropanoids [10,11,12,13,14]. However, there have been no reports on inositols from *S. capsicoides*.

In this study, we further explored the active component from the leaves of *S. capsicoides*, and eight inositol derivatives were isolated for the first time using different chromatographic techniques. The in vitro activity study found that the crude extract of the leaves of *S. capsicoides* exhibit anti-inflammatory activity [8], and at the same time, inositols have a beneficial effect on LPS-induced inflammatory responses in microglial cells [15]. We were inspired to evaluate the in vitro anti-inflammatory activity of isolated compounds using LPS-induced microglia.

## 2. Results and Discussion

The 70% ethanol extract from the leaves of *S. capsicoides* obtained hitherto unreported solsurinositols A–H (**1**–**8**), separated by chromatography (Figure 1), the spectrogram information can see Electronic Appendix A (Appendix A).

Compound **1** was obtained as a colorless solid. Its molecular formula, C_22_H_36_O_10_, was determined by positive HR-ESI-MS at m/z 478.2656 [M + NH_4_]^+^ (calcd. 478.2652) with five degrees of unsaturation. The ^13^C NMR and DEPT spectra (Table 1) showed signals at *δ*_C_ 72.5, 71.0, 69.8, 72.3, 72.3, and 73.9 for six oxygenated carbon atoms; four sets of carbonyl signals at *δ*_C_ 171.2, 177.2, 177.5, and 177.8; three methenyl carbons at *δ*_C_ 42.5, 42.6, and 35.1; two methylene carbons at *δ*_C_ 27.8 and 27.9; seven methyl groups at *δ*_C_ 11.9, 11.9, 17.0, 17.3, 19.1, 19.3, and 20.6. The ^1^H NMR spectrum (Table 2) showed six oxygenated protons at *δ*_H_ 5.34 (1H, t, *J* = 10.0 Hz), 5.06 (1H, dd, *J* = 10.0, 2.8 Hz), 5.56 (1H, t, *J* = 2.8 Hz), 4.89 (1H, dd, *J* = 10.0, 2.8 Hz), 3.84 (1H, t, *J* = 10.0 Hz), and 3.54 (1H, t, *J* = 10.0 Hz); seven methyl groups at *δ*_H_ 0.89 (3H, t, *J* = 7.4 Hz), 0.98 (3H, t, *J* = 7.4 Hz), 1.12 (3H, dd, *J* = 12.2, 7.0 Hz), 1.20 (3H, d, *J* = 7.0 Hz), 1.13 (3H, d, *J* = 7.0 Hz), 1.13 (3H, d, *J* = 7.0 Hz), and 1.90 (3H, s); three methine protons; two methylene protons. Compared with the literature (compound “Myo-Inositol pentabenzoate and Myo-inositol”) [16,17], the above data identified the inositol ring structure, and the difference here was that the new compound showed substituent groups. 

In the ^1^H-^1^H COSY (Figure 2) and HMBC spectra, the cross-peaks of H-1/H-2/H-3/H-4/H-5/H-6 were deduced to be one cyclohexane structure. The proton signals at *δ*_H_ 1.90 (3H, s) correlated with the carbon signal at *δ*_C_ 171.2, suggesting that there was one acetate moiety (Ac); the proton signals at *δ*_H_ 1.13 (3H, d, *J* = 7.0 Hz) and 1.13 (3H, d, *J* = 7.0 Hz) correlated with the carbon signal at *δ*_C_ 177.8, with the cross-peaks of H_iBu-2_/H_iBu-3_/H_iBu-4_ suggesting that there was one isobutyryloxy moiety (iBu); the proton signals at *δ*_H_ 2.41 (1H, q, *J* = 7.0 Hz), 1.46 (1H, m), 1.64 (1H, m), and 1.12 (3H, dd, *J* = 12.2, 7.0 Hz) correlated with the carbon signal at *δ*_C_ 177.5, with the cross-peaks of H_2MB-3_/H_2MB-2_/H_2MB-4_/H_2MB-5_ suggesting that there was a 2-methylbutyroyloxy moiety (2MB). The NMR data were compared with those in the literature, showing that these fragments were in accordance with the findings of previous studies [18,19]. The proton signals at *δ*_H_ 2.52 (1H, overlap), 1.57 (1H, m), 1.72 (1H, m), and 1.20 (1H, d, *J* = 7.0 Hz) correlated with the carbon signal at *δ*_C_ 177.2, with the cross-peaks of H_2MB-3′_/H_2MB-2′_/H_2MB-4′_/H_2MB-5′_ suggesting that there was a 2-methyl butyroyloxy moiety, 2MB. The correlation from H-1 to C-2MB indicated that the 2MB was attached to C-1; the correlation from H-3 to C-2MB′ indicated that the 2MB′ was attached to C-3; the correlation from H-2 to C-Ac indicated that the Ac was attached to C-2; the correlation from H-4 to C-iBu indicated that the iBu was attached to C-4. Through comparison with the literature [20], we found that the coupling patterns of six oxygenated methine protons in **1** included two axial/axial and axial/equatorial couplings with small *J* values of 2.8 Hz, as well as trans-diaxial couplings with large *J* values of 10.0. The coupling constants of **1** indicated the presence of the following: H-1 axial at *δ*_H_ 5.34 (1H, t, *J* = 10.0 Hz), H-2 axial at *δ*_H_ 5.06 (1H, dd, *J* = 10.0, 2.8 Hz), H-3 equatorial at *δ*_H_ 5.56 (1H, t, *J* = 2.8 Hz), H-4 axial at *δ*_H_ 4.89 (1H, dd, *J* = 10.0, 2.8 Hz), H-5 axial at *δ*_H_ 3.84 (1H, t, *J* = 10.0 Hz), and H-6 axial at *δ*_H_ 3.54 (1H, t, *J* = 10.0 Hz). Furthermore, the NOESY correlations of H-2/H-4, H-6, and the coupling constant *δ*_H_ 5.56 (1H, t, *J* = 2.8 Hz) of H-3 demonstrated that the relative configuration of H-2/H-4, H-6, and H-3 was determined as β. Therefore, compound **1** was determined to be myoinositol-1,3-(2-methylbutyrate)-2-acetate-4-isobutyryloxy, and was named solsurinositol A.

Compound **2** was obtained as a colorless solid. Its molecular formula, C_21_H_34_O_10_, was determined by positive HR-ESI-MS at *m*/*z* 464.2498 [M + NH_4_]^+^ (calcd. 464.2496) with five degrees of unsaturation. The downfield chemical shifts of H-1 at *δ*_H_ 5.33 (1H, t, *J* = 10 Hz), H-2 at *δ*_H_ 5.08 (1H, dd, *J* = 10.0, 2.8 Hz), H-3 at *δ*_H_ 5.55 (1H, t, *J* = 2.8 Hz), H-4 at *δ*_H_ 4.89 (1H, overlap), H-5 at *δ*_H_ 3.84 (1H, t, *J* = 10 Hz), and H-6 at *δ*_H_ 3.56 (1H, t, *J* = 10 Hz) also suggested that **2** possesses a 1,2,3,4-tetra-substituted inositol ring as in **1**. Through ^1^H-^1^H COSY and HMBC (Figure 3) spectra, the correlation from H-1 to C-iBu indicated that the iBu was attached to C-1; the correlation from H-2 to C-iBu′ indicated that the iBu′ was attached to C-2; the correlation from H-3 to C-2MB indicated that the 2MB was attached to C-3; the correlation from H-4 to C-Ac indicated that the Ac was attached to C-4. The relative configuration was determined by the same method as that used for **1**. The NOESY correlations of H-1/H-5 demonstrated that the relative configuration of H-1/H-5 was determined as α. Therefore, compound **2** was determined to be myoinositol-1,2-isobutyryloxy-3-(2-methylbutyrate)-4-acetate, and was named solsurinositol B.

Compound **3** was obtained as a colorless solid. Its molecular formula, C_25_H_40_O_11_, was determined by positive HR-ESI-MS at *m*/*z* 534.2916 [M + NH_4_]^+^ (calcd. 534.2914) with six degrees of unsaturation. The NMR data of 3 were similar to those of 1 (Table 1 and Table 2), which meant that they had the same cyclohexane structure. The main difference was the substituent patterns of the inositol ring. Through ^1^H-^1^H COSY and HMBC (Figure 4) spectra, the correlation from H-1 to C-Ac indicated that the Ac was attached to C-1; the correlation from H-2 to C-2MB indicated that the 2MB was attached to C-2; the correlation from H-3 to C-2MB′ indicated that the 2MB′ was attached to C-3; the correlation from H-4 to C-Ac′ indicated that the Ac′ was attached to C-4; the correlation from H-5 to C-2MB″ indicated that the 2MB″ was attached to C-5. The relative configuration was determined by the same method as that used for **1**. The configurations of H-1 at *δ*_H_ 5.41 (1H, td, *J* = 10.0, 5.2 Hz) and H-5 at *δ*_H_ 5.41 (1H, td, *J* = 10.0, 5.2 Hz) were defined by coupling constant as β. Therefore, compound **3** was determined to be myoinositol-1,4-acetate-2,3,5-(2-methylbutyrate), and was named solsurinositol C.

Compound **4** was obtained as a colorless solid. Its molecular formula, C_27_H_44_O_11_, was determined by positive HR-ESI-MS at *m*/*z* 562.3228 [M + NH_4_]^+^ (calcd. 562.3227) with six degrees of unsaturation. The NMR data of **4** (Table 1 and Table 2) were similar to those of **1**, which meant that they had the same cyclohexane structure. In ^1^H-^1^H COSY and HMBC, the correlation from H-1 to C-2MB indicated that the 2MB was attached to C-1; the correlation from H-2 to C-2MB′ indicated that the 2MB′ was attached to C-2; the correlation from H-3 to C-2MB″ indicated that the 2MB″ was attached to C-3; the correlation from H-4 to C-Ac indicated that the Ac was attached to C-4; the correlation from H-5 to C-iBu indicated that the iBu was attached to C-5. The NMR data were compared with those in the literature, showing that these fragments were consistent with those that had been found previously [21]. The relative configuration was determined by the same method as that used for **1**. The configurations of H-1 and H-5 were defined by coupling constant *δ*_H_ at 5.43 (1H, t, *J* = 10.0 Hz) and 5.38 (1H, t, *J* = 10.0 Hz) as β. Therefore, compound **4** was myoinositol-1,2,3-(2-methylbutyrate)-4-acetate-5-isobutyryloxy, and was named solsurinositol D.

Compound **5** was obtained as a colorless solid. Its molecular formula, C_21_H_36_O_9_, was determined by positive HR-ESI-MS at *m*/*z* 433.2424 [M + H]^+^ (calcd. 433.2438) with four degrees of unsaturation. ^1^H and ^13^C NMR spectra were close to compound **1** with the presence of one cyclohexane structure. The main difference was the substituent pattern and number. In ^1^H-^1^H COSY and HMBC (Figure 4) spectra, the correlation from H-2 to C-2MB indicated that the 2MB was attached to C-2; the correlation from H-3 to C-2MB′ indicated that the 2MB′ was attached to C-3; the correlation from H-6 to C-2MB″ indicated that the 2MB″ was attached to C-6 [22]. The coupling patterns of six oxygenated methine protons were close to **1**. Therefore, compound **5** was myoinositol-2,3,6-(2-methylbutyrate), and was named solsurinositol E.

Compound **6** was obtained as a colorless solid. Its molecular formula, C_18_H_30_O_9_, was determined by positive HR-ESI-MS at *m*/*z* 408.2228 [M + NH_4_]^+^ (calcd. 408.2234) with four degrees of unsaturation. The NMR data of **6** were similar to those of **1**, which meant that they had a similar cyclohexane structure. In ^1^H-^1^H COSY and HMBC (Figure 5), the correlation from H-2 to C-2MB indicated that the 2MB was attached to C-2; the correlation from H-3 to C-2MB′ indicated that the 2MB′ was attached to C-3; the correlation from H-4 to C-Ac indicated that the Ac was attached to C-4. The NOESY (Figure 5) correlations demonstrated that the relative configurations of H-2/H-4 and H-6 were determined as β. Based on the above ^1^H NMR and ^13^C NMR spectrum data, the structure of compound **6** was similar to that of 1β-acetate 2α-methybutanoate 3α-methybutanoate, except for the relative configuration [21]. Through the analysis of the compound, it was found that the configuration of the fourth position of the compound was opposite. Therefore, compound **6** was myoinositol-2,3-(2-methylbutyrate)-4-acetate, and was named solsurinositol F.

Compound **7** was obtained as a colorless solid. Its molecular formula, C_18_H_36_O_8_, was determined by positive HR-ESI-MS at *m*/*z* 408.2249 [M + NH_4_]^+^ (calcd. 408.2228) with four degrees of unsaturation. The ^1^H and ^13^C NMR spectra were close to compound **1** with the presence of one six-membered ring structure. The main difference was the side connections. In ^1^H-^1^H COSY and HMBC (Figure 6) spectra, the correlation from H-1 to C-2MB indicated that the 2MB was attached to C-1; the correlation from H-2 to C-Ac indicated that the Ac was attached to C-2; the correlation from H-3 to C-2MB′ indicated that the 2MB′ was attached to C-3 [23]. The coupling patterns of six oxygenated methine protons were close to those of **1**. Therefore, compound **7** was myoinositol-1,3-(2-methylbutyrate)-2-acetate, and was named solsurinositol G.

Compound **8** was obtained as a colorless solid. Its molecular formula, C_21_H_36_O_9_, was determined by positive HR-ESI-MS at *m*/*z* 450.2715 [M + NH_4_]^+^ (calcd. 450.2703) with four degrees of unsaturation. ^1^H and ^13^C NMR spectra were close to compound **1** with the presence of one cyclohexane structure. The main difference was the substituent pattern and number. In ^1^H-^1^H COSY and HMBC (Figure 6) spectra, the correlation from H-2 to C-2MB indicated that the 2MB was attached to C-2; the correlation from H-3 to C-2MB′ indicated that the 2MB′ was attached to C-3; the correlation from H-4 to C-2MB″ indicated that the 2MB″ was attached to C-4. The coupling patterns of six oxygenated methine protons were close to **1**. Therefore, compound **8** was myoinositol-2,3,6-(2-methylbutyrate), and was named solsurinositol H.

Using the CCK8 method, the in vitro cytotoxicity of all isolated compounds on BV2 cells was evaluated in vitro. The results are listed in Table 3. Compounds **1**–**8** were not significantly cytotoxic to BV2 cells (IC_50_ > 100 µM).

Inositols were shown to have anti-inflammatory activity in the literature, which was tested against LPS-induced NO (nitric oxide) production in RAW 264.7 macrophages and BV2 microglia [22,24]. Our study also supported previous results. In this study, isolated compounds **1**−**8** were examined for their inhibition of NO production in LPS-stimulated BV2 cells (Table 4). Compounds **3** and **4** showed very weak NO-production-inhibition activities, with IC_50_ > 100 µM. Compounds **1**, **2**, **6**, and **8** exhibited inhibitory activities, with IC_50_ values of 24.27 ± 1.82, 31.66 ± 2.71, 23.31 ± 0.74, and 31.03 ± 0.92 µM, respectively. Compounds **5** (IC_50_ = 11.21 ± 0.14 µM) and **7** (IC_50_ = 14.5 ± 1.22 µM) had good anti-inflammatory activity in vitro. Inositol was reported to be effective in treating central nervous system disorders such as depression, Alzheimer’s disease, and obsessive–compulsive disorder in [25,26]. Our experimental results showed that compounds **5** and **7** were promising for the treatment of neuroinflammation. Although it was reported that Solanum has anti-inflammatory effects, and the steroids and flavonoids were considered to be the active components [27,28,29], our study found that inositol derivatives isolated from this plant also had potential anti-inflammatory effects. 

## 3. Materials and Methods

### 3.1. General Experimental Procedures

HR-ESI-MS spectra were selected on a Thermo Orbitrap Fusion Lumos Tribrid mass spectrometer (Thermo Fisher Scientific, Waltham, MA, USA). NMR spectra were used in a Bruker DPX-600 spectrometer (Bruker Company Ltd, Karlsruhe, Germany) with TMS as an internal standard. Optical rotation was measured on the Jasco P-2000 digital polarizer. Preparative HPLC was measured by LC-20AR pump and RID-20A detector (flow rate: 5 mL/min) by a C18 column (5 μm, 20 × 250 mm, 5 mL/min, Shimadzu, Japan). Column chromatography was carried out on silica gel (80–100 mesh and 200–300 mesh, Shanghai Titan Scientific Co, Shanghai, China) and ODS (Octadecylsilyl) (YMC Company Ltd., Kyoto, Japan).

### 3.2. Plant Material

The leaves of *Solanum capsicoides* were collected from Kunming (Yunnan Province) in 2020. We harvested in August when the leaves were growing most vigorously. The plant sample was identified by Professor Rui-Feng Fan of the Heilongjiang University of Chinese Medicine, and its certificate specimen (No. 20200826) is now stored in the Key Laboratory of Basic and Application Research of Beiyao, Ministry of Education of Heilongjiang University of Chinese Medicine.

### 3.3. Extraction and Isolation

The dried leaves of *S. capsicoides* (20.0 kg) were refluxed 3 times with 70% EtOH for 2 hours each time, to obtain a crude extract (6.0 kg) after removal of the solvent under vacuum. The 3.0 kg crude extract was eluted through the resin of HP-20 to obtain water (749.0 g), 30% ethanol (343.0 g), and 95% ethanol extracts (423.0 g).

The 95% EtOH extract was fractionated by a silica gel column (200–300 mesh) using CH_2_Cl_2_/EtOH in a gradient (10:0 to 0:10) to yield eight fractions (Fr. A−H). Fr. C (23.9 g) was separated by MCI (Middle Chromatogram Isolated) chromatography, and eluted with CH_3_OH/H_2_O (1:9–10:0). Through TLC analysis, 22 sub-fractions were obtained. Fr. C-14 was purified by preparative HPLC (MeOH/H_2_O 70%) to obtain compound **1** (19.9 mg, t_R_ = 24.8 min) and compound **3** (9.3 mg t_R_ = 33.27 min). Fr. C-16 was purified by preparative HPLC (MeOH/H_2_O 75%) to obtain compound **4** (40.4 mg, t_R_ = 28.6 min). The Fr. D (21.6 g) was dealt with MCI chromatography and elution with CH_3_OH/H_2_O (1:9–10:0) in sequence to obtain 20 sub-fractions. Fr. D-12 was purified by preparative HPLC (MeOH/H_2_O 68%) to obtain compound **2** (14.4 mg, t_R_ = 45.3 min) and compound **5** (24.0 mg, t_R_ = 50.7 min). The Fr. E (35.0 g) was dealt with MCI chromatography and eluted with CH_3_OH/H_2_O (1:9–10:0) in sequence to obtain 3 sub-fractions. Fr. E-1 (4.8 g) was dealt with ODS chromatography and eluted with CH_3_OH/H_2_O (1:9–10:0) in sequence to obtain 50 sub-fractions. Fr. E1-35 was purified by preparative HPLC (MeOH/H_2_O 65%) to obtain compound **7** (11.9 mg, t_R_ = 20.6 min). Fr. E1-36 was purified by preparative HPLC (MeOH/H2O 65%) to obtain compound **6** (10.2 mg, t_R_ = 31.1 min). Fr. E-2 (12.3 g) was dealt with ODS chromatography and eluted with CH_3_OH/H_2_O (1:9–10:0) in sequence to obtain 26 sub-fractions. Fr. E2-17 was purified by preparative HPLC (MeOH/H2O 65%) to obtain compound **8** (12.2 mg, t_R_ = 15.9 min).

Solsurinositol A (**1**)

Colorless solid; ${\left[\alpha \right]}_{D}^{22}+5.2$ (c 1.00, MeOH); ^1^H NMR (methanol-*d*_4_, 600 MHz): (Table 1); ^13^C NMR (methanol-*d*_4_, 150 MHz): (Table 2); HR-ESI-MS *m*/*z* 478.2656 [M + NH_4_]^+^ (calcd. for C_22_H_40_NO_10_, 478.2652).

Solsurinositol B (**2**)

Colorless solid; ${\left[\alpha \right]}_{D}^{22}+44.3$ (c 1.00, MeOH); ^1^H NMR (methanol-*d*_4_, 600 MHz): (Table 1); ^13^C NMR (methanol-*d*_4_, 150 MHz): (Table 2); HR-ESI-MS *m*/*z* 464.2498 [M + NH_4_]^+^ (calcd. for C_21_H_38_NO_10_, 464.2496). 

Solsurinositol C (**3**)

Colorless solid; ${\left[\alpha \right]}_{D}^{22}+10.6$ (c 1.00, MeOH); ^1^H NMR (methanol-*d*_4_, 600 MHz): (Table 1); ^13^C NMR (methanol-*d*_4_, 150 MHz): (Table 2); HR-ESI-MS *m*/*z* 534.2916 [M + NH_4_]^+^ (calcd. for C_25_H_44_NO_11_, 534.2914). 

Solsurinositol D (**4**)

Colorless solid; ${\left[\alpha \right]}_{D}^{22}+37.8$ (c 1.00, MeOH); ^1^H NMR (methanol-*d*_4_, 600 MHz): (Table 1); ^13^C NMR (methanol-*d*_4_, 150 MHz): (Table 2); HR-ESI-MS *m*/*z* 562.3228 [M + NH_4_]^+^ (calcd. for C_27_H_48_NO_11_, 562.3227). 

Solsurinositol E (**5**)

Colorless solid; ${\left[\alpha \right]}_{D}^{22}+36.4$ (c 1.00, MeOH); ^1^H NMR (methanol-*d*_4_, 600 MHz): (Table 1); ^13^C NMR (methanol-*d*_4_, 150 MHz): (Table 2); HR-ESI-MS *m*/*z* 433.2424 [M + H]^+^ (calcd. for C_21_H_37_O_9_, 433.2438). 

Solsurinositol F (**6**)

Colorless solid; ${\left[\alpha \right]}_{D}^{22}+5.58$ (c 1.00, MeOH); ^1^H NMR (methanol-*d*_4_, 600 MHz): (Table 1); ^13^C NMR (methanol-*d*_4_, 150 MHz): (Table 2); HR-ESI-MS *m*/*z* 408.2228 [M + NH_4_]^+^ (calcd. for C_18_H_34_NO_9_, 408.2234). 

Solsurinositol G (**7**)

Colorless solid; ${\left[\alpha \right]}_{D}^{22}+16.1$ (c 1.00, MeOH); ^1^H NMR (methanol-*d*_4_, 600 MHz): (Table 1); ^13^C NMR (methanol-*d*_4_, 150 MHz): (Table 2); HR-ESI-MS *m*/*z* 408.2249 [M + NH_4_]^+^ (calcd. for C_18_H_34_NO_9_, 408.2234).

Solsurinositol H (**8**)

Colorless solid; ${\left[\alpha \right]}_{D}^{22}+2.76$ (c 1.00, MeOH); ^1^H NMR (methanol-*d*_4_, 600 MHz): (Table 1); ^13^C NMR (methanol-*d*_4_, 150 MHz): (Table 2); HR-ESI-MS *m*/*z* 450.2715 [M + NH_4_]^+^ (calcd. for C_21_H_40_NO_9_, 450.2703).

### 3.4. Bioactive Activity

The cytotoxicity assay was performed with a modified CCK8 method. First, BV2 microglial cells were cultured with 96-well microplates (10^4^ cells/well in 100 L medium) for 24 h. Then, the cells were treated with different concentrations (5, 10, 30, 50, 100, 200 μM) of compounds. After the cells were incubated for 24 h, added 10 μL CCK8 solution to each well and incubated for 2 h in a 5% CO_2_ humidifier at 37 ℃. The absorbance was recorded at 450 nm with a microplate reader (BioTek Company, Winooski, VT, USA), repeating three times, and its IC_50_ value was calculated.

BV2 microglial cells were cultured in DMEM containing 10% fetal bovine serum, 100 IU/mL penicillin, and 100 μg/mL streptomycin at 37 ℃ with 5% CO_2_. Then, the cells were placed in 96-well microplates and treated with different compounds at **1**–**8** concentrations (5, 10, 30, 50, 100, 200 μM) for 1 h. Next, the cells were treated with 1 μg/mL LPS for 24 h to detect the NO (nitric oxide) levels in culture supernatants. After the Griess reaction (incubated at 37 ℃ with 5% CO_2_ for 0.5 h), the absorbance of cells was measured with a microplate reader (BioTek Company, Winooski, VT, USA) at 450 nm, repeated three times, and its IC_50_ value was calculated.

## 4. Conclusions

In summary, eight new inositol derivatives were isolated from *S. capsicoides*, named solsurinositols A–H (**1**–**8**). Their structures were determined by extensive spectroscopic analysis, including 1D, 2D NMR, and HR-ESI-MS. The inflammation model was established by LPS-induced BV2 microglia, and the anti-inflammatory effects of all compounds (**1**–**8**) were evaluated. Except for compounds **3** and **4**, the compounds had varying degrees of anti-inflammatory effects, with IC_50_ values ranging from 11 to 35 µM. Compounds **5** (IC_50_ = 11.21 ± 0.14 µM) and **7** (IC_50_ = 14.5 ± 1.22 µM) had potential anti-inflammatory activity. Inhibition of inflammation caused by hyperactivation of microglia is one of the effective strategies for treating neurodegenerative diseases. Therefore, compounds **5** and **7** are promising for the treatment of neuroinflammation. It is hoped that our study can provide a new direction for the study and application of *S. capsicoides*. We look forward to investigating the exact mechanism of action in further studies.

## Figures and Tables

**Figure 1 molecules-27-06063-f001:**
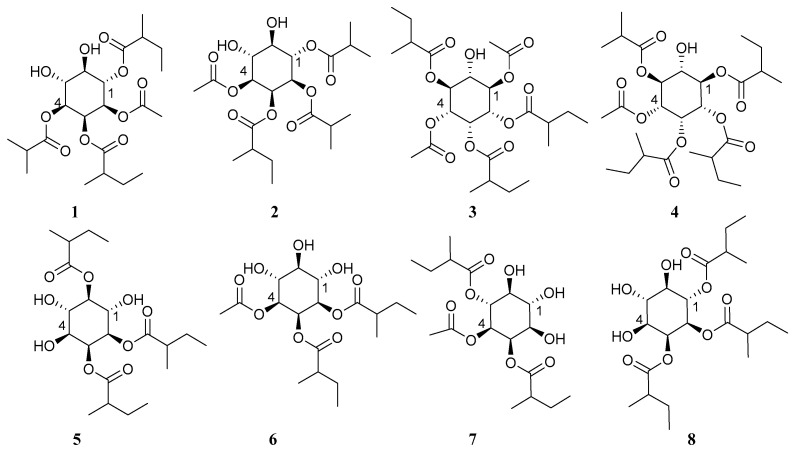
Chemical structures of compounds **1**–**8**.

**Figure 2 molecules-27-06063-f002:**
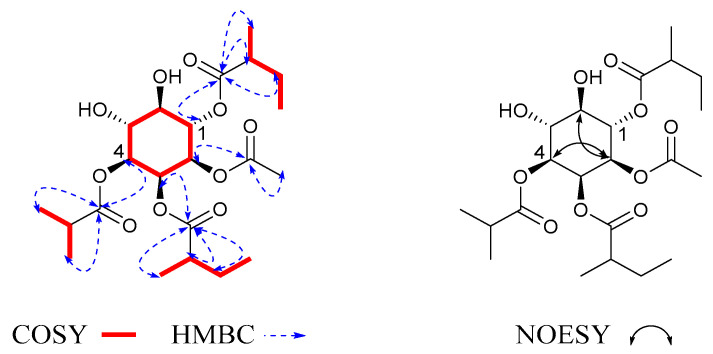
^1^H-^1^H COSY, key HMBC and NOE correlations of compound **1**.

**Figure 3 molecules-27-06063-f003:**
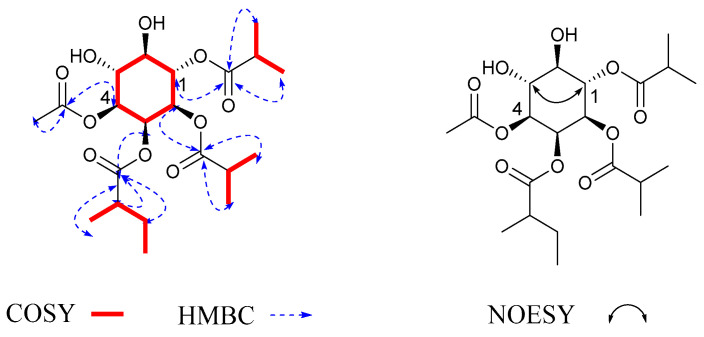
^1^H-^1^H COSY, key HMBC and NOE correlations of compound **2**.

**Figure 4 molecules-27-06063-f004:**
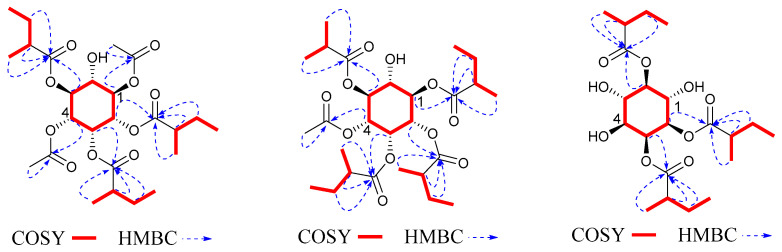
^1^H-^1^H COSY, key HMBC correlations of compounds **3**, **4**, and **5**.

**Figure 5 molecules-27-06063-f005:**
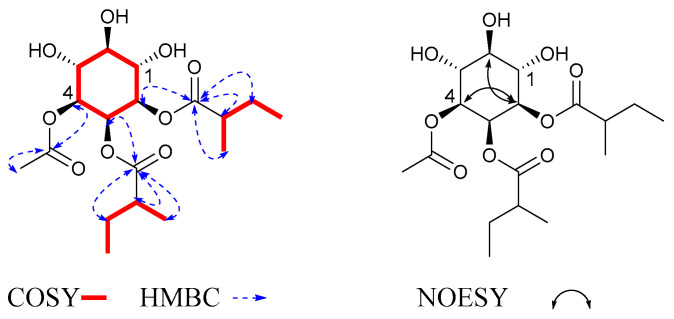
^1^H-^1^H COSY, key HMBC, and NOE correlations of compound **6**.

**Figure 6 molecules-27-06063-f006:**
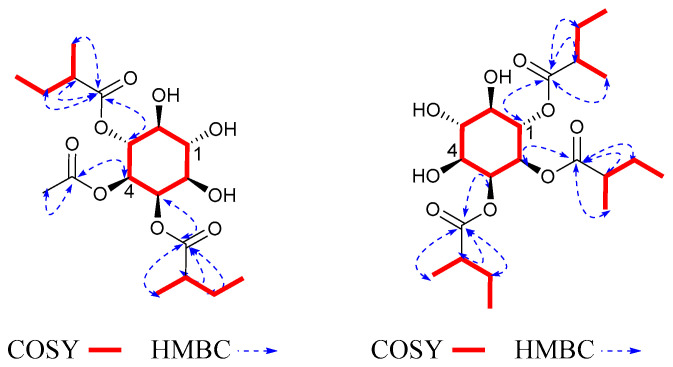
^1^H-^1^H COSY, key HMBC correlations of compounds **7** and **8**.

**Table 1 molecules-27-06063-t001:** The 600 MHz 1H-NMR data of compounds **1**–**8** (in CD3OD δ in ppm, *J* in Hz).

Position	1	2	3	4	5	6	7	8
1	5.34 (t, 10.0)	5.33 (t, 10.0)	5.41 (td, 10.0, 5.2)	5.43 (t, 10.0)	3.85 (t, 10.1)	3.78 (t, 9.7)	5.30 (t, 10.0)	3.44 (t, 10.0)
2	5.06 (dd, 10.0, 2.8)	5.08 (dd, 10.0, 2.8)	5.15 (dt, 10.0, 2.8)	5.16 (ddd, 13.4, 10.0, 2.8)	4.83 (overlap)	4.83 (overlap)	4.92 (dd, 10.0, 2.8)	5.50 (t, 2.8)
3	5.56 (t, 2.8)	5.55 (t, 2.8)	5.60 (t, 2.8)	5.60 (t, 2.8)	5.51 (t, 2.5)	5.53 (t, 2.8)	5.48 (t, 2.8)	4.96 (dd, 10.0, 2.8)
4	4.89 (dd, 10.0, 2.8)	4.89 (overlap)	5.15 (dt, 10.0, 2.8)	5.16 (ddd, 13.4, 10.0, 2.8)	3.69 (overlap)	4.81 (overlap)	3.44 (t, 10.0)	5.32 (t, 10.0)
5	3.84 (t, 10.0)	3.84 (t, 10.0)	5.41 (td, 10.0, 5.2)	5.38 (t, 10.0)	3.69 (overlap)	3.75 (t, 9.7)	3.67 (overlap)	3.66 (overlap)
6	3.54 (t, 10.0)	3.56 (t, 10.0)	3.79 (t, 10.0)	3.79 (t, 10.0)	4.88 (overlap)	3.36 (t, 9.7)	3.67 (overlap)	3.66 (overlap)
2MB-2	2.41 (q, 7.0)	2.51 (m)	2.27 (m)	2.39 (m)	2.33 (m)	2.36 (m)	2.52 (m)	2.37 (q, 7.0)
2MB-3	1.46 (m) 1.64 (m)	1.71 (m)	1.43 (m) 1.61(m)	1.47 (m) 1.68 (m)	1.41 (m) 1.70 (overlap)	1.42 (m) 1.63 (m)	1.60 (overlap) 1.74 (m)	1.46 (m) 1.67 (m)
2MB-4	0.89 (t, 7.4)	0.98 (t, 7.4)	0.84 (t, 7.4)	0.92 (t, 7.4)	0.96 (t, 7.4)	0.85 (t, 7.4)	0.99 (t, 7.4)	0.92 (t, 7.5)
2MB-5	1.12 (dd, 12.2, 7.0)	1.20 (d, 7.0)	1.05 (d, 7.0)	1.10 (d, 7.0)	1.18 (d, 7.0)	1.11 (d, 7.0)	1.17 (d, 7.0)	1.21 (d, 7.0)
2MB-2′	2.52 (overlap)		2.55 (m)	2.26 (m)	2.46 (m)	2.45 (m)	2.41 (m)	2.22 (q, 7.0)
2MB-3′	1.57 (m) 1.72 (m)		1.61 (m) 1.75 (m)	1.39 (m) 1.59 (m)	1.51 (m) 1.70 (overlap)	1.53 (m) 1.70 (m)	1.45 (m) 1.60 (overlap)	1.37 (m) 1.61 (overlap)
2MB-4′	0.98 (t, 7.4)		1.00 (t, 7.4)	0.83 (t, 7.4)	0.96 (t, 7.4)	0.97 (t, 7.4)	0.88 (t, 7.4)	0.86 (t, 7.4)
2MB-5′	1.20 (d, 7.0)		1.22 (d, 7.0)	1.04 (d, 7.0)	1.18 (d, 7.0)	1.17 (d, 7.0)	1.12 (d, 7.0)	1.04 (d, 7.0)
2MB-2″			2.41 (m)	2.56 (m)	2.46 (m)			2.48 (q, 7.0)
2MB-3″			1.43 (m) 1.61 (m)	1.59 (m) 1.76 (m)	1.51 (m) 1.70 (overlap)			1.52 (m) 1.74 (overlap)
2MB-4″			0.89 (t, 7.4)	1.01 (t, 7.4)	0.95 (t, 7.4)			0.94 (t, 7.4)
2MB-5″			1.13 (d, 7.0)	1.22 (d, 7.0)	1.18 (d, 7.0)			1.10 (d, 7.0)
Ac-2	1.90 (s)	2.00 (s)	2.04 (s)	1.91 (s)		1.99 (s)	1.89 (s)	
Ac-2′			1.91 (s)					
iBu-2	2.52 (overlap)	2.56 (m)		2.56 (m)				
iBu-3	1.13 (d, 7.0)	1.12 (d, 7.0)		1.13 (t, 7.1)				
iBu-4	1.13 (d, 7.0)	1.15 (d, 7.0)		1.13 (t, 7.1)				
iBu-2′		2.41 (m)						
iBu-3′		1.06 (dd, 7.0, 1.2)						
iBu-4′		1.06 (dd, 7.0, 1.2)						

**Table 2 molecules-27-06063-t002:** The 150 MHz ^13^C-NMR data **1**–**8** (in CD_3_OD; *δ* in ppm, *J* in Hz).

Position	1	2	3	4	5	6	7	8
1	72.5	72.8	73.3	72.7	70.6	72.1	72.3	74.2
2	71.0	70.6	70.2	70.4	73.3	72.4	71.7	73.0
3	69.8	69.8	69.6	69.7	72.4	69.9	72.6	71.5
4	72.3	72.7	70.8	70.8	71.1	73.1	73.9	72.5
5	72.3	72.2	72.3	72.8	72.6	72.0	74.5	74.6
6	73.9	73.7	71.6	71.6	76.6	75.9	70.6	70.9
2MB-1	177.5	177.2	176.6	177.2	177.6	177.6	178.2	177.5
2MB-2	42.5	42.6	42.4	42.2	42.4	42.2	42.4	42.2
2MB-3	27.8	27.9	27.5	27.7	27.5	27.7	28.0	27.7
2MB-4	11.9	12.0	12.0	11.8	11.9	11.8	11.8	11.8
2MB-5	17.0	17.4	17.0	17.0	17.0	16.8	16.8	17.4
2MB-1′	177.2		177.0	176.6	177.4	177.4	177.9	176.9
2MB-2′	42.6		42.6	42.1	42.5	42.6	42.3	42.2
2MB-3′	27.9		27.9	27.5	27.7	27.9	27.8	27.3
2MB-4′	11.9		12.0	11.8	11.9	12.0	11.8	11.8
2MB-5′	17.3		17.4	16.7	16.9	17.4	16.9	16.7
2MB-1″			177.2	177.0	178.2			177.3
2MB-2″			42.5	42.6	42.5			42.6
2MB-3″			27.8	27.9	27.9			27.7
2MB-4″			11.9	12.0	11.9			12.0
2MB-5″			17.1	17.3	17.4			17.0
Ac-1	171.2	171.9	171.7	171.1		172.1	171.8	
Ac-2	20.6	20.7	20.9	20.5		20.8	20.8	
Ac-1′			171.1					
Ac-2′			20.6					
iBu-1	177.8	177.9		177.8				
iBu-2	35.1	35.2		35.2				
iBu-3	19.1	19.2		19.0				
iBu-4	19.3	19.5		19.6				
iBu-1′		177.1						
iBu-2′		35.1						
iBu-3′		19.0						
iBu-4′		19.1						

**Table 3 molecules-27-06063-t003:** Cytotoxic activities of compounds **1**–**8**.

No.	IC_50_ (µM)	No.	IC_50_ (µM)
1	>100	5	>100
2	>100	6	>100
3	>100	7	>100
4	>100	8	>100

**Table 4 molecules-27-06063-t004:** Inhibitory effects of compounds **1**–**8** on NO in LPS-induced BV-2.

No.	IC_50_ (µM)	No.	IC_50_ (µM)
1	24.27 ± 1.82	6	23.31 ± 0.74
2	31.66 ± 2.71	7	14.50 ± 1.22
3	>100	8	31.03 ± 0.92
4	>100	NMMA	1.67 ± 0.24
5	11.21 ± 0.14		

## Data Availability

Data are contained within the article and Appendix A.

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
