# Peer review of "Inositol Derivatives with Anti-Inflammatory Activity from Leaves of Solanum capsicoides Allioni"

_molecules, 2022, doi:10.3390/molecules27186063_

Round 1

Reviewer 1 Report

The manuscript aimed to screen S. capsicoides extracts for the presence of inositols and evaluate the anti-inflammatory activity of the isolated compounds. The study is interesting; however the rationale of the study should be clearer and the results on anti-inflammatory activity should be discussed properly.

Abstract: The abstract should be revised, including more details on methods and results.

Introduction: should be improved

Line 23 – Make clearer the rationale of the study. Detail the importance of inositols in human reproduction and metabolism and why it is important to find sources of this compound in nature. What is the rationale to screen the presence of inositols in S. capsicoides and why is this relevant?

Material and Methods

Include the period of the year when the leaves of S. capsicoides were collected.

Why was 70% EtOH selected for extraction?

Bioactive activity: Include the parameters of classification for toxicity. How authors classified “compounds 1-8 as not significantly cytotoxic to BV2 cells”?

In item 3.4 – include the full denomination for “NO levels”.

Discussion – The results on anti-inflammatory activity should be discussed properly. What are the potential applications of the obtained results?

English revision is needed. Some parts of the text are confusing and difficult to understand.

Reviewer 2 Report

This paper describes the isolation and structure elucidation of 8 new natural products from the leaves of Solanum capsicoides, a plant that has been used in traditional medicine. The natural products were purified by various chromatographic techniques, and characterized by high-resolution mass spectrometry and by the standard suite of 1D and 2D NMR experiments. The natural products are all inositols, sharing the same relative stereochemistry on the ring (i.e. myo-inositol) but with different acyl groups (e.g. acetate, isobutyrate, 2-methylbutyrate) attached to various of the hydroxyl groups. Inositols have previously been shown to have anti-inflammatory activity, and some of the new natural products in this paper are found to possess this property. Overall I believe that this paper makes a significant contribution to knowledge.

Major issues that need to be addressed before publication is warranted:

·      A discussion of the absolute stereochemistry needs to be included. Although the myo-inositol scaffold is achiral, the non-symmetrical positioning of the acyl groups renders each of these 8 natural products chiral, and the nonzero optical rotations suggests that they are non-racemic.

·      Chiral HPLC traces should be included, to provide additional evidence of whether the natural products are enantiopure or scalemic.

·      Were any efforts made to obtain crystal structures? That might have helped to assign the absolute stereochemistry. (All compounds were obtained in decent quantity – tens of milligrams – and all are solids.) This should be discussed.

·      It would be more logical to show a single general structure in Figure 1, rather than having compounds 3 and 4 arbitrarily viewed from the opposite face compared to the others.

·      It would be helpful to the reader if the inositol structure could be always drawn in a consistent orientation in Figures 1-6. Currently it is sometimes rotated in the plane of the page (e.g. Figure 6), or viewed from the opposite face (e.g. Figure 4), which is needlessly confusing.

·      Figures 2, 3, 5: opposite absolute configurations are drawn in the two representations of compounds 1, 2 and 6. This should be fixed.

Minor / typographical issues:

·      Figure 1: use superscript rather than subscript numerals for R1, R2, etc

·      Figure 1: in the legend that explains the structures of MB, Ac, iBu, make it clearer where the point of attachment is to the rest of the molecule

·      Line 68: data not date

·      Line 141: close not closed

·      Captions for Figures 4 and 6: delete mention of NOE

·      Line 217: avoid abbreviation “m” for mesh (might get confused with meters)

·      Section 3.3: define the acronyms MCI, ODS

·      Figures 2, 5: re-write “2MB” substituent (not “BM2”)
